# Leveraging Large Language Models for Precision Monitoring of Chemotherapy-Induced Toxicities: A Pilot Study with Expert Comparisons and Future Directions

**DOI:** 10.3390/cancers16162830

**Published:** 2024-08-12

**Authors:** Oskitz Ruiz Sarrias, María Purificación Martínez del Prado, María Ángeles Sala Gonzalez, Josune Azcuna Sagarduy, Pablo Casado Cuesta, Covadonga Figaredo Berjano, Elena Galve-Calvo, Borja López de San Vicente Hernández, María López-Santillán, Maitane Nuño Escolástico, Laura Sánchez Togneri, Laura Sande Sardina, María Teresa Pérez Hoyos, María Teresa Abad Villar, Maialen Zabalza Zudaire, Onintza Sayar Beristain

**Affiliations:** 1Department of Mathematics and Statistic, NNBi 2020 SL, 31110 Noain, Navarra, Spain; maialen.zabalza@nnbi.es; 2Medical Oncology Service, Basurto University Hospital, OSI Bilbao-Basurto, Osakidetza, 48013 Bilbao, Biscay, Spain; mariapurificacion.martinezdelprado@osakidetza.eus (M.P.M.d.P.); marian.salagonzalez@osakidetza.eus (M.Á.S.G.); josune.azcunasagarduy@osakidetza.eus (J.A.S.); pablo.casadocuesta@osakidetza.eus (P.C.C.); covadonga.figaredoberjano@osakidetza.eus (C.F.B.); elena.galvecalvo@osakidetza.eus (E.G.-C.); borja.lopezdesanvicentehernandez@osakidetza.eus (B.L.d.S.V.H.); maria.lopezsantillan@osakidetza.eus (M.L.-S.); maitane.nunoescolastico@osakidetza.eus (M.N.E.); laura.sancheztogneri@osakidetza.eus (L.S.T.); laura.sandesardina@osakidetza.eus (L.S.S.); nani.perezhoyos@osakidetza.eus (M.T.P.H.); mariateresa.abadvillar@osakidetza.eus (M.T.A.V.)

**Keywords:** Large Language Models, artificial intelligence, oncology, clinical practice, chemotherapy, subjective toxicities, medical oncology, patient monitoring

## Abstract

**Simple Summary:**

This study evaluated the ability of Large Language Models (LLMs) to classify subjective toxicities from chemotherapy by comparing them with expert oncologists. Using fictitious cases, it was demonstrated that LLMs can achieve accuracy similar to that of oncologists in general toxicity categories, although they need improvement in specific categories. LLMs show great potential for enhancing patient monitoring and reducing the workload of doctors. Future research should focus on training LLMs specifically for medical tasks and validating these findings with real patients, always ensuring accuracy and ethical data management.

**Abstract:**

Introduction: Large Language Models (LLMs), such as the GPT model family from OpenAI, have demonstrated transformative potential across various fields, especially in medicine. These models can understand and generate contextual text, adapting to new tasks without specific training. This versatility can revolutionize clinical practices by enhancing documentation, patient interaction, and decision-making processes. In oncology, LLMs offer the potential to significantly improve patient care through the continuous monitoring of chemotherapy-induced toxicities, which is a task that is often unmanageable for human resources alone. However, existing research has not sufficiently explored the accuracy of LLMs in identifying and assessing subjective toxicities based on patient descriptions. This study aims to fill this gap by evaluating the ability of LLMs to accurately classify these toxicities, facilitating personalized and continuous patient care. Methods: This comparative pilot study assessed the ability of an LLM to classify subjective toxicities from chemotherapy. Thirteen oncologists evaluated 30 fictitious cases created using expert knowledge and OpenAI’s GPT-4. These evaluations, based on the CTCAE v.5 criteria, were compared to those of a contextualized LLM model. Metrics such as mode and mean of responses were used to gauge consensus. The accuracy of the LLM was analyzed in both general and specific toxicity categories, considering types of errors and false alarms. The study’s results are intended to justify further research involving real patients. Results: The study revealed significant variability in oncologists’ evaluations due to the lack of interaction with fictitious patients. The LLM model achieved an accuracy of 85.7% in general categories and 64.6% in specific categories using mean evaluations with mild errors at 96.4% and severe errors at 3.6%. False alarms occurred in 3% of cases. When comparing the LLM’s performance to that of expert oncologists, individual accuracy ranged from 66.7% to 89.2% for general categories and 57.0% to 76.0% for specific categories. The 95% confidence intervals for the median accuracy of oncologists were 81.9% to 86.9% for general categories and 67.6% to 75.6% for specific categories. These benchmarks highlight the LLM’s potential to achieve expert-level performance in classifying chemotherapy-induced toxicities. Discussion: The findings demonstrate that LLMs can classify subjective toxicities from chemotherapy with accuracy comparable to expert oncologists. The LLM achieved 85.7% accuracy in general categories and 64.6% in specific categories. While the model’s general category performance falls within expert ranges, specific category accuracy requires improvement. The study’s limitations include the use of fictitious cases, lack of patient interaction, and reliance on audio transcriptions. Nevertheless, LLMs show significant potential for enhancing patient monitoring and reducing oncologists’ workload. Future research should focus on the specific training of LLMs for medical tasks, conducting studies with real patients, implementing interactive evaluations, expanding sample sizes, and ensuring robustness and generalization in diverse clinical settings. Conclusions: This study concludes that LLMs can classify subjective toxicities from chemotherapy with accuracy comparable to expert oncologists. The LLM’s performance in general toxicity categories is within the expert range, but there is room for improvement in specific categories. LLMs have the potential to enhance patient monitoring, enable early interventions, and reduce severe complications, improving care quality and efficiency. Future research should involve specific training of LLMs, validation with real patients, and the incorporation of interactive capabilities for real-time patient interactions. Ethical considerations, including data accuracy, transparency, and privacy, are crucial for the safe integration of LLMs into clinical practice.

## 1. Introduction

Large Language Models (LLMs) have emerged as a revolutionary technology with broad and diverse applications across numerous fields, including medicine. LLMs, such as the GPT model family from OpenAI [1], are artificial intelligence systems trained on vast amounts of textual data, enabling them to understand and generate text coherently and contextually. These models can learn and understand new tasks without specific training, allowing them to adapt and solve a wide variety of problems. This versatility and ability to function in different situations and environments represent the most significant revolution of LLMs. In the medical field, these models have the potential to transform various clinical practices by providing support in tasks ranging from documentation to direct patient interaction and assistance in clinical decision making.

The use of Large Language Models in medicine is still under development with the primary goal of improving the accuracy, efficiency, and quality of medical practice. These models have the capability to process large volumes of clinical data and medical literature, providing healthcare professionals with relevant and updated information quickly. In oncology, where continuous and precise patient monitoring is crucial, LLMs can play a fundamental role. The best support for oncology treatments is personalized care, which is available 24 h a day, 7 days a week. However, due to obvious limitations, healthcare providers cannot offer this continuous monitoring to all patients.

Focusing on the monitoring of toxicities in patients undergoing chemotherapy treatments, daily real-time follow-up, or follow-up when the patient deems it necessary, would significantly improve care and help prevent risk situations, such as critical toxicity phases requiring hospitalization or modification of the treatment plan. However, it is unfeasible to allocate human resources to conduct such exhaustive monitoring for all patients. An artificial intelligence system, on the other hand, could perform this task by evaluating all patients in real time. Combined with a well-implemented alert system, it could become a monitoring system that significantly improves the quality of care. This system would allow human and technical resources to be focused where and when they are needed, optimizing the response to patients’ needs in each situation.

Specifically, the study aimed to evaluate whether an LLM would be able to correctly identify the presence and severity of subjective toxicities, that is, those that do not require analytical tests but rather the patient’s subjective description. The research focuses on determining whether a contextualized LLM could interpret these subjective descriptions in a manner like professional oncologists with clinical experience. The ability of an LLM to accurately understand and evaluate these subjective toxicities could revolutionize patient monitoring, allowing for personalized and continuous care without the need for additional human resources, improving the quality of care and reducing risks associated with treatment toxicities.

### 1.1. Applications of LLMs in Cancer Care

The integration of Large Language Models (LLMs) in cancer care has been explored in several recent studies, showing promising results. A comprehensive study reviewed the latest advancements in the use of LLMs in the healthcare sector, highlighting their transformative impact in various medical domains, including cancer [2]. These models improve the accuracy of diagnosis and treatment, providing support to both doctors and patients by handling various types of medical data and integrating complex information to offer more precise diagnoses and treatments.

LLMs have proven useful in monitoring toxicities in cancer patients by processing patient self-reported data on their symptoms and facilitating the early identification of toxicities to allow timely interventions [3]. Additionally, they have been used to manage electronic health records (EHRs), keeping data up to date and efficiently organized. In tumor boards, where critical treatment decisions are made, LLMs can provide quick and accurate summaries of patient cases, supporting medical teams in making informed decisions. However, while they can offer adequate information on the screening or management of specific solid tumors, they also exhibit a significant rate of errors and a tendency to provide outdated data, making a rigorous verification process by experts essential [3].

In monitoring radiotherapy-related toxicities, a pilot study investigated the use of GPT-4 for monitoring toxicity in prostate cancer treatments, comparing a summary method and a chatbot interface. Radiation oncologists preferred the summary method for its accuracy and potential for adoption, demonstrating that both methods saved time [4]. However, not all studies have been favorable. An analysis of the reliability of ChatGPT in responses related to radiation oncology revealed that 34.1% of the responses contained incorrect information, 26.2% lacked essential context, and only 39.7% were correct and complete, highlighting the need for a rigorous verification process [5]. Another study evaluated the accuracy and comprehensiveness of ChatGPT in domains related to radiation oncology, finding that it frequently generated inaccurate or incomplete responses with only 39.7% of patient-centered questions being considered correct and complete [6].

In summary, although LLMs have proven to be valuable tools in cancer care by providing support in toxicity monitoring, medical record management, and decision making, their effective implementation requires a careful approach. A rigorous verification process and the development of clear guidelines are necessary to maximize the benefits of these technologies and minimize the risks associated with incorrect or incomplete information. As LLMs continue to evolve, their integration into clinical practice promises to significantly improve the quality of cancer care, provided that appropriate safeguards are implemented to ensure their accuracy and reliability. Additionally, it is crucial to address ethical and technical challenges, such as managing diverse medical data, ensuring algorithm transparency, avoiding bias, and protecting data privacy [2,4,5,6,7,8,9,10].

### 1.2. State of the Art in Toxicity Monitoring

Toxicity monitoring in cancer patients has significantly evolved with the use of advanced technologies and the collection of patient-reported outcomes (PROs). One study evaluated the feasibility of lung cancer patients undergoing chemotherapy self-reporting their toxicity symptoms using an online platform, finding high participation and satisfaction, demonstrating that it is a feasible long-term strategy [11]. Another study reviewed the role of artificial intelligence in integrating electronic health records with patient-generated health data, identifying benefits such as patient empowerment, improved patient–provider relationships, and reduced time and costs of clinical visits [12].

The routine collection of PROs in oncology has improved patient–provider communication and patient satisfaction, although evidence on its impact on patient management and health outcomes is limited. Nevertheless, there is growing support for the collection of PROs to enable more patient-centered care [13]. The incorporation of electronic PRO (ePRO) assessments significantly enhances the quality of care, although it faces challenges in integrating PRO data into clinical workflows and electronic medical record systems [14].

A randomized controlled trial showed that symptom self-reporting improves health-related quality of life (HRQL), with fewer emergency admissions and hospitalizations, and longer chemotherapy duration [15]. The development of the PRO-CTCAE by the National Cancer Institute has improved the documentation of symptomatic adverse events in cancer clinical trials, allowing patients to report the frequency, severity, and interference of adverse events in their daily activities [16].

The evaluation of an online platform for cancer patients to self-report their toxicity symptoms during chemotherapy showed high levels of use and acceptance, improving safety and satisfaction with care by providing real-time data and alerts for clinicians [17]. Finally, technological interventions such as AMTRA have been introduced to monitor treatment-related toxicity in a standardized manner, showing high compliance and improvements in communication between patients and providers [18].

These studies highlight the potential and challenges of implementing technology-based and patient-reported data systems for toxicity monitoring, underscoring the importance of continuing to develop and refine these tools to improve cancer care [19,20,21,22].

### 1.3. Overview and Paper Structure

In this work, we propose utilizing Large Language Models to detect and classify subjective toxicities in fictitious patients undergoing chemotherapy. This study serves as a foundation for future research involving real patients to validate and refine the application of LLMs in clinical oncology settings.

The rest of the paper is arranged as follows:Methods: describes the study design, including the creation of fictitious cases, the evaluation process by expert oncologists, and the contextualization of the LLM model for classifying subjective toxicities.Results: presents the comparative analysis between the evaluations performed by oncologists and the LLM model, detailing accuracy metrics and types of errors.Discussion: interprets the findings, highlighting the potential and limitations of LLMs in clinical practice and suggesting directions for future research.Conclusions: summarizes the key findings and their implications for clinical oncology, emphasizing the need for specific training of LLMs and further studies with real patients.

## 2. Methods

### 2.1. Study Design

#### 2.1.1. Study Type

This study was designed as a comparative pilot analysis aimed at evaluating the ability of a language model to classify subjective toxicities arising from chemotherapy treatments. The pilot nature of the study addresses the need to establish the feasibility and accuracy of LLMs in this context before advancing to research phases involving real patients.

#### 2.1.2. Study Objectives

The study has three primary objectives:Evaluate the ability of an LLM to classify subjective toxicities in patients undergoing chemotherapy.Assess the feasibility of using fictitious cases as a basis to justify future studies with real patients.Evaluate the accuracy of a contextualized LLM model without having been specifically trained for the task of classifying subjective toxicities.

#### 2.1.3. General Approach Description

The general approach of the study involved the creation of fictitious cases, the generation of audio recordings of these cases, the evaluation of toxicities by experienced oncologists, and the comparison of these evaluations with those conducted by a contextualized LLM model. The LLM used in this study was a version of OpenAI’s GPT-4, which was specifically contextualized for the task of toxicity classification. This model was chosen because it had the best benchmarks in terms of performance and accuracy at the time of conducting the experiment [23,24,25,26,27].

### 2.2. Ethical Considerations

In this research, no real patients were involved, so it was not necessary to obtain informed consent from patients. This aspect is relevant as it was a pilot study aimed at evaluating the feasibility of the proposed tools before advancing to subsequent phases involving the participation of real patients.

The purpose of this pilot project was to obtain preliminary results that could justify and support the implementation of future studies with real patients. The initial goal was to determine the capability and accuracy of the LLM model in classifying subjective toxicities using fictitious cases, so that the obtained results could be promising enough to proceed to the next phase of research. This future phase would include obtaining informed consent and direct evaluation with patients, always ensuring compliance with applicable ethical and legal standards.

### 2.3. Participants

#### 2.3.1. Participant Description

The study included thirteen oncologists with current clinical experience in evaluating toxicities arising from oncology treatments. These professionals were selected based on their availability and expertise in the study area. The participating oncologists are part of the Medical Oncology department at Basurto University Hospital, belonging to Osakidetza, the Basque Health Service.

#### 2.3.2. Number of Participating Oncologists

Initially, the participation of fifteen oncologists was planned with eight assigned to the first group and seven to the second group. The oncologists were randomly assigned to the groups to ensure an equitable and representative distribution. However, due to the lack of response from two oncologists, thirteen ultimately participated with eight in the first group and five in the second. Each group received a set of 15 fictitious cases for evaluation. This division into two groups was made at the request of the oncologists, as having all thirteen oncologists classify the 30 cases would have taken twice as much time. By splitting them into two groups, each oncologist only needed to classify 15 cases, thus making the task more manageable and time-efficient while maintaining the necessary equity and representativeness for the evaluation of the fictitious cases.

#### 2.3.3. Inclusion and Exclusion Criteria

We used the following inclusion criteria for the participating oncologists:Current clinical practice in medical oncology.Familiarity with the use of the CTCAE version 5.0 [28].Availability to participate in the entire study.

Oncologists were excluded if they:Could not complete the evaluation of all assigned cases.

### 2.4. Fictitious Cases

#### 2.4.1. Creation of Fictitious Cases

The creation of the fictitious cases was carried out independently from the group of medical oncologists who later evaluated them, ensuring impartiality in the subsequent evaluations. Expert knowledge was used to guarantee the realism and clinical relevance of the cases. A total of 30 fictitious cases were generated to simulate typical toxicological scenarios in patients undergoing chemotherapy treatments. The number of cases was determined based on the estimated time each oncologist would need to evaluate them, as evaluating a larger number of cases would have been unfeasible for this pilot study.

#### 2.4.2. Process of Generating 30 Expert-Knowledge-Based Fictitious Cases

The process of generating the fictitious cases included several key stages. First, based on expert knowledge, detailed clinical scenarios were designed, including precise descriptions of patient symptoms and experiences. To avoid potential biases, we ensured that the representation of toxicities was proportionate to their real-world incidence both in terms of presence and severity. The designed scenarios were then reviewed and validated to ensure their realism and clinical appropriateness. Once the toxicological profiles of the fictitious patients were created, 30 plausible cases were generated, ensuring a diverse and realistic representation of the toxicities under study. For each profile, the GPT-4 model was used to create a realistic narrative for each fictitious patient, naturally expressing the toxicities they experienced and interweaving personal and daily life details to add realism to the cases. In Appendix A, we define the prompt used to generate these transcriptions; since the transcription was generated in Spanish, the prompt is also in Spanish. Additionally, an example of a transcription is included in the same annex. Both the prompt and the transcription in the annex have been translated into English.

After generating the narrative texts, OpenAI’s Text-to-Speech model was used to create an audio recording for each case. Finally, a further evaluation was conducted to ensure the realism of the 30 generated audios, considering the described toxicities, the personal stories told, and the manner of expression. The generated audios have a duration of approximately 3 min each. The audios were generated in Spanish. An example of a generated audio in Spanish is included in the Appendix A.

To ensure the realism of the fictitious cases, updated and accurate information about the side effects and common toxicities of chemotherapy treatments was used. Additionally, detailed and specific descriptions were achieved that reflect real patient experiences, ensuring that the cases were representative of authentic clinical situations.

### 2.5. Toxicity Chart

#### Evaluation of Toxicity According to CTCAE v.5

The classification of toxicities in oncology patients is conducted using the Common Terminology Criteria for Adverse Events of the National Cancer Institute, version 5.0. This system provides a standardized scale to assess the severity of adverse events (AEs) that may occur as a result of medical treatment. The scale ranges from Grade 0 to Grade 5, each associated with specific clinical descriptions:Grade 0 (None): Absence of the evaluated toxicity.Grade 1 (Mild): Mild or asymptomatic symptoms that do not require medical intervention. Patients can continue with their daily activities without significant interruptions.Grade 2 (Moderate): Moderate symptoms that may require minimal, local, or non-invasive intervention. Patients experience limitations in Instrumental Activities of Daily Living (IADLs) but can manage them with some difficulty.Grade 3 (Severe): Severe, medically significant, and potentially disabling symptoms. They may require hospitalization or the prolongation of hospital stay, limiting the patient’s ability to perform self-care Activities of Daily Living (ADLs).Grade 4 (Life-threatening): Adverse events with potentially life-threatening consequences that require urgent medical intervention. They represent an immediate threat to the patient’s life.Grade 5 (Death): Adverse events resulting in the patient’s death.

This categorization allows for a systematic and uniform evaluation of AEs, facilitating the proper monitoring and management of toxicities in patients undergoing oncology treatment. In Appendix A we provide a detailed list of the evaluated toxicities and the specific grading criteria according to CTCAE v.5.

### 2.6. Expert Evaluation

The oncologists evaluated the subjective toxicities by listening to the audios of the fictitious patients and using the CTCAE v.5.0 as a reference guide. Each oncologist classified the reported toxicities in the audios, determining the severity of each symptom according to the scales established in the CTCAE.

### 2.7. Contextualization of the LLM Model

For this study, the version of GPT-4 developed by OpenAI trained with data up to December 2023 was selected. This model was chosen due to its advanced ability to understand and generate natural language in a wide variety of contexts and being the most powerful model available at the time the study was conducted.

#### Process of Contextualizing the Model for Classifying Subjective Toxicities

For this study, a customized GPT model was generated [29] and specifically contextualized for the task of classifying subjective toxicities. This customized GPT model utilizes the GPT-4 architecture trained with data up to December 2023. The contextualization process included providing the model with all relevant information about the CTCAE v.5 classification method and its criteria. The nature of the toxicities, the descriptions of the grades, and how to interpret the symptoms reported by patients were explained in detail. The customized GPT model was indexed with, among other things, the CTCAE v.5 guide to use as a reference. It is important to note that at no point was the model shown a single real classification case as a reference. In this way, the model’s ability to apply the CTCAE v.5 criteria was evaluated solely based on the contextualization and the information provided. In Appendix A, the process of contextualizing this GPT model is explained in detail.

### 2.8. Analysis of Results

Since each case had been analyzed by 8 or 5 oncologists, depending on the group to which it was assigned, it was necessary to adopt a metric that could reflect the consensus among the evaluations. For this purpose, two main metrics were used: the mode and the mean of the responses. The mode represents the most assigned classification by the oncologists, while the mean provides an average value of the given classifications.

Each case was evaluated by the contextualized model a total of 10 times. This number of evaluations was chosen to introduce variability into the model’s responses, allowing for a more robust and comprehensive assessment of its performance. By incorporating this level of variability, we can ensure that the model’s final response is not an outlier or overly influenced by a single evaluation. To obtain the final response of the model for each case, both the mode and the mean of these 10 evaluations were taken, thus ensuring the stability and representativeness of the responses generated by the model.

The comparison of the oncologists’ results and the LLM model was conducted through a detailed analysis of the subjective toxicity classifications in the fictitious cases. Two main approaches were used: evaluating accuracy in general categories and specific categories.
Accuracy General Categories: This was defined as the agreement between the severity classifications of toxicities by the LLM model and the classifications by the oncologists, grouping the toxicities into general categories. These categories were divided into “no toxicity” (0), “mild toxicities” (1–2), and “severe toxicities” (3–4). Accuracy was calculated by comparing whether the model’s classification was within the same general category as the mode of the oncologists.Accuracy Specific Categories: The model’s accuracy was evaluated in terms of exact agreement with the oncologists’ specific classifications for each toxicity without grouping them.

#### 2.8.1. Types of Errors (Mild and Severe)


Mild Errors: Mild errors are those where the model classified a toxicity with a higher grade than indicated by the oncologists. For example, if a patient presents with a mild taste alteration (grade 1) and the model classifies it as moderate (grade 2). In a real clinical setting, mild errors could lead to more conservative management, involving additional tests. While this may increase the workload for healthcare staff and associated costs, it does not directly compromise patient safety.Severe Errors: Severe errors are defined as those where the model classified a toxicity with a lower grade than indicated by the oncologists: for example, if a patient presents with severe diarrhea (grade 3) and the model classifies it as moderate (grade 1). These errors can have a more significant negative impact on patient management. Underestimating the severity of a toxicity could delay appropriate treatment, increasing the risk of serious complications. For example, not correctly identifying severe diarrhea could lead to severe patient dehydration and avoidable hospitalizations.


#### 2.8.2. False Alarm Analysis

False alarms were defined as situations where the model classified a toxicity as severe (3 or 4) when the mean of the oncologists did not consider it as such.

## 3. Results

This section presents the results obtained from comparing the evaluations of subjective toxicities performed by expert oncologists and the classifications generated by a contextualized Large Language Model (LLM). The results are divided into several subsections that address the dispersion in expert evaluation, the evaluation of the mode and mean of the classifications, as well as a comparative analysis between these approaches. This analysis is fundamental for understanding the accuracy and reliability of the LLM model in classifying toxicities arising from chemotherapy treatment.

### 3.1. Dispersion in Expert Evaluation

Considerable variability was observed in the oncologists’ evaluations when classifying the subjective toxicities arising from chemotherapy treatments. Below are the ordered values of the mean entropy by type of toxicity from the oncologists’ responses. Higher entropy indicates greater variability in the responses given by the oncologists (see Table 1).

The main reason for this variability in evaluations is that the oncologists worked with closed audios of fictitious patients, which did not allow for interaction or the possibility of asking additional questions. In a real clinical setting, oncologists can delve deeper into the patients’ responses, clarify doubts, and obtain additional information that allows for a more precise and uniform evaluation. However, in this study, the lack of direct interaction limited their ability to correctly interpret the reported symptoms, resulting in greater variability in toxicity classifications. This methodological limitation highlights the importance of interaction in clinical evaluation and underscores one of the challenges of using models based on pre-recorded audios.

### 3.2. Mode Evaluation

Table 2 summarizes the results of the LLM model in classifying toxicities by comparing two evaluations: the mode and mean of the LLM model against the mode and mean of the oncologists’ evaluations.

Accuracy in general categories refers to the model’s ability to classify toxicities into general severity categories—no toxicity (0), mild toxicities (1–2), and severe toxicities (3–4)—in a manner that matches the oncologists’ classifications. Accuracy in specific categories measures the model’s precision in exactly matching the classification of each toxicity grade (0, 1, 2, 3, 4) with the oncologists’ evaluations.

Errors are classified as mild or severe based on their potential impact on patient management. Mild errors occur when the model classifies a toxicity with a higher grade than indicated by the oncologists, while severe errors occur when the model classifies a toxicity with a lower grade, underestimating the severity of the patient’s condition.

False alarms are defined as situations where the model classifies a toxicity as severe (grade 3 or 4), while the mean of the oncologists’ evaluations does not consider the toxicity to be severe.

To compare these results with what could be considered expert-level outcomes, the following approach was taken. Each oncologist’s individual results were compared to the rest of their peers who evaluated the same cases. This comparison aimed to determine the percentage of times an individual oncologist’s responses matched those of the other oncologists who assessed the same cases.

First, by calculating each oncologist’s accuracy percentages, ranges of accuracy percentages for individual oncologists were obtained for both accuracy in general categories and accuracy in specific categories, for both the mean and the mode. This helped define an overall range of maximum and minimum percentages that could be considered expert level.

Second, a confidence interval for the central tendency of the oncologists’ results was quantified to see if the LLM results fell within these intervals. The median was used to construct the confidence intervals, as the oncologists’ results did not meet normality criteria. The confidence intervals were constructed with a 95% confidence level.

Table 3 presents the range of accuracy percentages for individual oncologists in classifying toxicities. The ranges are shown for both the mode and mean evaluations across general and specific categories. The ranges provided help define what could be considered expert-level performance.

Table 4 presents the 95% confidence intervals for the median accuracy of oncologists in classifying toxicities. The intervals are shown for both the mode and mean evaluations across general and specific categories. These confidence intervals help determine the range within which the true median accuracy of oncologists’ evaluations is expected to fall, providing a benchmark for expert-level performance.

## 4. Discussion

### 4.1. Summary of Main Findings

This study has demonstrated that Large Language Models (LLMs) can classify subjective toxicities arising from chemotherapy treatments with a level of accuracy comparable to that of expert oncologists. The key results show that the LLM model achieved an accuracy of 81.5% in the classification of general toxicity categories and 64.4% in the classification of specific toxicity categories when comparing modes. When comparing the means, the respective results were 85.7% and 64.6%. These findings are particularly relevant in the context of the study’s objectives, which were to evaluate the ability of LLMs to classify subjective toxicities, assess the feasibility of working with fictitious cases, and evaluate the accuracy of a contextualized model without specific training.

The results of this study were further analyzed against the ranges and confidence intervals defined for expert-level performance. As shown in Table 3, the LLM model’s accuracy in general categories falls well within the range of 72.5% to 89.0% for the mode and 66.7% to 89.2% for the mean, indicating that the model’s performance is on par with that of individual oncologists. However, in specific categories, the LLM’s accuracy (64.4% for the mode and 64.6% for the mean) is on the lower end of the expert range, which spans from 64.2% to 80.0% for the mode and 57.0% to 76.0% for the mean.

In Table 4, the 95% confidence intervals for the median accuracy of oncologists provide a benchmark for assessing the LLM model’s results. The model’s accuracy in general categories (81.5% for the mode and 85.7% for the mean) comfortably falls within the confidence intervals (81.3% to 87.9% for the mode and 81.9% to 86.9% for the mean). However, for specific categories, the LLM’s accuracy (64.4% for the mode and 64.6% for the mean) does not fall within the confidence intervals (72.9% to 77.2% for the mode and 67.6% to 75.6% for the mean), indicating that while the LLM performs well, there is room for improvement to reach the median level of expert oncologists.

### 4.2. Interpretation of the Results

The LLM model’s results were compared with evaluations performed by expert oncologists. The accuracy achieved by the model in classifying toxicities falls within the range of accuracy of oncologists, as shown in Table 3. These results are significant because the model was not specifically trained for this task but was only contextualized to address toxicity classification problems.

The variability in oncologists’ evaluations, reflected in the entropy of their classifications, highlights the inherent difficulty in evaluating subjective toxicities. This variability may be further influenced by the subjective nature of the toxicities and the lack of direct interaction with patients in this study. Despite this variability, the LLM model showed consistent performance within the expert range for general categories. However, for specific categories, the model’s accuracy did not fall within the confidence intervals for the median accuracy of oncologists, as shown in Table 4. This indicates that while the LLM model performs well and is comparable to expert oncologists in general categorization, there is room for improvement in the exact classification of specific toxicity grades, underscoring its potential and the need for further refinement.

### 4.3. Errors

In the error analysis, two main types of errors were identified: mild errors and severe errors. Mild errors, representing 96% of the total errors, occurred when the model classified a toxicity with a higher grade than indicated by the oncologists. These errors are less concerning, as they could result in additional interventions that, while unnecessary, do not pose a risk to the patient.

On the other hand, severe errors, which represent only 4% of the total errors, occurred when the model classified a toxicity with a lower grade than indicated by the oncologists. These errors are more critical, as they could lead to underestimating the severity of the patient’s condition, potentially resulting in insufficient care.

### 4.4. Limitations

One of the main limitations of the study is the use of fictitious cases and pre-recorded audios instead of real interactions with patients. The lack of direct interaction limits the oncologists’ ability to obtain additional information and clarify doubts, which is crucial for accurate clinical evaluation. This method may have contributed to the observed variability in the oncologists’ classifications and, by extension, in the LLM model’s classifications.

Additionally, the LLM model used in this study was contextualized but not specifically trained for the task of toxicity classification. This means that although the model demonstrated promising performance, its accuracy could be significantly improved with specific training and greater contextualization with relevant clinical data.

Another important limitation is that the study did not include an interactive component, where the LLM model could engage in a dynamic conversation with the patient. In a real clinical setting, the ability to ask additional questions and adjust evaluations based on the patient’s responses is crucial for accurate assessment. The inclusion of an interactive component could potentially improve the model’s accuracy and reduce variability in classifications.

A further significant limitation is that the LLM model performed classifications solely based on audio transcriptions. This results in the loss of information about how the patient speaks, which is fundamental for making accurate toxicity classifications. Voice tone, pauses, and other speech characteristics can provide crucial clues about the severity of symptoms that are not captured in textual transcriptions. This limitation could be addressed using new models such as the upcoming GPT-4o [30], which will be capable of directly processing audio and comprehending its full complexity.

Finally, although the results obtained are promising, additional studies with real patients are necessary to validate these findings. The use of real data will allow for a more accurate evaluation of the LLM model’s effectiveness and reliability in a clinical setting as well as its impact on daily medical practice.

In summary, although the study presents several limitations, the results indicate high potential for the use of LLMs in classifying subjective toxicities in cancer patients. Implementing these models in clinical practice could significantly improve the efficiency and accuracy of patient monitoring provided the identified limitations are addressed and specific training and evaluations in real clinical settings are conducted.

### 4.5. Clinical and Practical Relevance

The use of Large Language Models in monitoring toxicities in cancer patients has significant potential to transform clinical practice. LLMs can process large volumes of textual data and provide accurate and rapid assessments, which is crucial in monitoring patients undergoing chemotherapy treatments. By effectively identifying and classifying toxicities, these models can facilitate early interventions and reduce the risk of severe complications.

One of the most notable benefits of implementing LLMs in clinical practice is the reduction in oncologists’ workload. LLMs can take on repetitive and time-consuming tasks, allowing oncologists to focus on more complex aspects of patient care. This not only improves the efficiency of patient monitoring but also optimizes the use of available medical resources.

Furthermore, the ability of LLMs to identify toxicities early and accurately can lead to significant improvements in the quality of care. The early detection of side effects allows for timely adjustments in treatments, enhancing patients’ tolerance to chemotherapy and ultimately improving their quality of life. The continuous and personalized monitoring that LLMs can offer is a significant advancement toward more patient-centered care.

### 4.6. Future Implications

Despite the promising results, this study highlights the need to train LLM models specifically for medical tasks. Specialized training can significantly enhance the accuracy and reliability of LLMs in classifying toxicities, further reducing the rate of severe errors and false alarms. The potential for implementing LLMs in real clinical environments is considerable. These models can be integrated into health management systems to provide continuous support in patient monitoring, assisting healthcare professionals in making informed decisions based on updated and accurate data.

For future research, it is crucial to conduct studies with real patients to validate the findings obtained from fictitious cases. Additionally, interactive evaluation, where LLMs can engage in dynamic conversations with patients, could further improve the accuracy of assessments. Proposals for future research include the following:Studies with Real Patients: Conduct studies with real patients to validate the accuracy and utility of the LLM model in a clinical setting. These studies should consider the AI system’s performance across different demographic groups, disease severity levels, and comorbidities.Specific Model Training: Develop and train LLM models specifically for the classification of oncological toxicities, using relevant clinical datasets. Tailoring the training data to include a wide variety of clinical scenarios will enhance the model’s accuracy and reliability.Interactive Evaluations: Implement evaluations where LLMs can interact directly with patients, adjusting their assessments in real time. Real-time interactions can provide additional context and clarification, leading to more precise assessments.Sample Expansion: Include a larger number of oncologists and cases to ensure the representativeness and robustness of the results. A more extensive sample size will enhance the generalizability of the findings.Robustness and Generalization: Ensure healthcare AI systems demonstrate robust performance across diverse and challenging scenarios, including variations in data quality, noise, missing data, and adversarial attacks. Robustness testing should evaluate the AI system’s performance under different conditions and assess its ability to generalize to unseen data and real-world clinical settings.

These future directions will not only improve the performance of LLMs but also facilitate their integration into clinical practice, optimizing the care of cancer patients and improving their health outcomes.

## 5. Conclusions

### 5.1. Summary of Key Findings

This study has shown that Large Language Models (LLMs) can classify subjective toxicities from chemotherapy with accuracy comparable to expert oncologists. The LLM’s performance in general toxicity categories (81.5% for mode and 85.7% for mean) falls within the expert range (72.5% to 89.0% for mode and 66.7% to 89.2% for mean) and within the confidence intervals for median accuracy (81.3% to 87.9% for mode and 81.9% to 86.9% for mean).

However, in specific toxicity categories, the LLM’s accuracy (64.4% for mode and 64.6% for mean) is at the lower end of the expert range (64.2% to 80.0% for mode and 57.0% to 76.0% for mean) and falls outside the confidence intervals for median accuracy (72.9% to 77.2% for mode and 67.6% to 75.6% for mean), indicating room for improvement in exact matches to expert classifications.

### 5.2. Study Impact

The results of this study are highly relevant for clinical practice and the monitoring of toxicities in cancer patients. The ability of LLMs to process large volumes of textual data and provide accurate and rapid assessments can revolutionize the way patient monitoring is conducted for those undergoing chemotherapy. By effectively identifying and classifying toxicities, these models enable early interventions and reduce the risk of severe complications, improving the quality of care and clinical process efficiency.

### 5.3. Future Research Directions

Considering the results obtained, several recommendations are proposed for the implementation of LLMs in the monitoring of cancer patients. First, it is crucial to train LLM models specifically for the task of classifying subjective toxicities. This specific training will improve accuracy and reduce the error rate, particularly severe errors.

Additionally, further studies with real patients are recommended to validate the findings obtained with fictitious cases. Evaluating the model’s effectiveness in a real clinical setting is essential to ensure its applicability and reliability in daily medical practice.

Finally, it is suggested to incorporate an interactive component in future versions of the LLM model, allowing it to interact directly with patients to adjust evaluations in real time. This interactive capability will not only improve the accuracy of evaluations but also facilitate more personalized and patient-centered care.

In summary, LLMs have great potential to improve the accuracy and efficiency of subjective toxicity evaluations in cancer patients. With specific training and additional validations in real clinical settings, these models can become valuable tools for healthcare professionals, significantly enhancing patient care and optimizing available medical resources.

### 5.4. Ethical Considerations

The implementation of LLMs in clinical practice must be accompanied by rigorous verification and ethical management of patient data. It is essential to ensure that the data provided by LLMs are accurate and up to date, minimizing the possibility of errors and misinformation.

Transparency in the algorithms used by LLMs is crucial to gaining the trust of healthcare professionals and patients. This includes the need to avoid biases in training data and in the responses generated by the models. Biases can lead to disparities in medical care and negatively affect certain patient groups.

Finally, data privacy protection is a critical consideration. LLMs handle sensitive patient information, and it is essential that this information is managed in accordance with regulations and best practices in information security. Data confidentiality must be prioritized to ensure that patients trust the use of these technologies in their medical care.

In conclusion, while LLMs have the potential to transform medical care, it is essential to address these ethical and technical challenges to ensure their safe and effective integration into clinical practice.

## Figures and Tables

**Table 1 cancers-16-02830-t001:** Mean entropy values by type of toxicity from the oncologists’ responses. Higher entropy indicates greater variability in the responses given by the oncologists, indicating the difficulty in uniformly evaluating these subjective toxicities.

Variable	Value
Anorexia	0.794
Depression	0.664
Taste alteration	0.637
Asthenia	0.598
Hand-foot syndrome	0.554
Alopecia	0.507
Nausea	0.503
Diarrhea	0.501
Peripheral neuropathy	0.484
Mucositis	0.473
Vomiting	0.466
Insomnia	0.446
Skin alteration	0.411
Fever	0.383
Dyspnea	0.357
Hyperhidrosis	0.323
Headache	0.288
Constipation	0.224
Abdominal pain	0.217
Conjunctivitis	0.168
Hematuria	0.137

**Table 2 cancers-16-02830-t002:** LLM model results in toxicity classification.

	Mode	Mean
Accuracy in General Categories	81.5%	85.7%
Accuracy in Specific Categories	64.4%	64.6%
Mild Errors/Severe Errors	96/4%	96.4/3.6%
False Alarms	8.9%	3%

**Table 3 cancers-16-02830-t003:** Range of accuracy percentages for oncologists.

	Mode	Mean
Accuracy in General Categories	72.5–89.0%	66.7–89.2%
Accuracy in Specific Categories	64.2–80.0%	57.0–76.0%

**Table 4 cancers-16-02830-t004:** Confidence intervals for the median accuracy of oncologists.

	Mode	Mean
Accuracy in General Categories	81.3–87.9%	81.9–86.9%
Accuracy in Specific Categories	72.9–77.2%	67.6–75.6%

## Data Availability

Restrictions apply to the datasets. The datasets presented in this article are not readily available because the data are part of an ongoing study.

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
