# Peer review of "Leveraging Large Language Models for Precision Monitoring of Chemotherapy-Induced Toxicities: A Pilot Study with Expert Comparisons and Future Directions"

_cancers, 2024, doi:10.3390/cancers16162830_

Round 1
Reviewer 1 Report
Comments and Suggestions for Authors
Summary
This pilot study uses LLMs to interrogate audio tapes that present toxicities caused by systemic treatments in cancer. The results demonstrate considerable variability in the results which limit the accuracy of the models.
Major Comments
1. The paper is too long for a report of research results. There is too much background material presented e.g. review on toxicities and patient-related outcomes (PROS). There is a standard format for reporting research results in cancer medical literature. Suggested format: Introduction (1-2 pages), Methods, Results, Discussion (Where the original objectives fulfilled? Discuss results including limitations. Next Steps, Conclusions). Currently the paper is 19 pages. This needs to be shortened by at least one third. (max 3500 words).
2. The methodology used, i.e. using audio of scenarios on toxicities, is innovative but in fact may have contributed to the downfall of the study. The initial goal was to use this method to create a very accurate reflection of the toxicity patients experience. But it introduced excess variability, both inter and intra observer. Hence, is the study actually a test of LLMs or a test of the design of the study.
3. This is a study of “agreement”. Because of many variables, e.g. number of oncologists, number of case scenarios, lack of a precise outcome, the analysis is many-layered. The help of a biostatistician is needed. It would be helpful for the reader if the authors walked the reader through an example of what was done (perhaps in a supplement). Figures would be helpful. Would it have been better to use a regression model for analysis.
4. To date much of the research comparing LLMs in clinical medicine have used surrogate case scenarios. In general, the results have shown that this form of AI is not ready for prime in clinical practice. The current study confirms this conclusion. What do the investigators plan to do next? Will they stick with audio vignettes approach and streamline them and reduce complex methodology or will they abandon this research.
Other comments
5. I have made multiple editorial comments directly on the PDF to help the authors with revising and shortening the paper. I will include this with my review.

The quality of English is fine.
The paper is too long and not in a correct format to report scientific results.
Reviewer 2 Report
Comments and Suggestions for Authors
This study evaluates the use of a Large Language Model (GPT-4) for classifying subjective toxicities in fictitious chemotherapy patients, comparing its accuracy to that of expert oncologists. The model achieved high accuracy in general categories (81.5%) and specific categories (64.4%). The findings suggest that LLMs can accurately classify chemotherapy-induced toxicities, with further research needed to validate these results in real patients. The study highlights the potential for LLMs to enhance patient monitoring and reduce severe complications in oncology care.
There are some notable strengths of the manuscripts,
1. Utilizes advanced AI technology (LLM) to address a critical need in oncology care, offering a novel method for monitoring chemotherapy-induced toxicities.
2. Demonstrates that the LLM can achieve accuracy levels comparable to expert oncologists, particularly in general toxicity categories.
3. Includes detailed analysis of model performance, error types, and comparison metrics, ensuring a thorough evaluation of the LLM's capabilities.
4. The approach can potentially be scaled to other areas of oncology and different medical conditions, broadening its applicability and impact.
Few Questions / Comments on the manuscript are as follows,
1. The study uses 30 fictitious cases for evaluation. It would be good if the authors good have provide more details on the process and criteria used to ensure the realism and clinical relevance of these cases?
2. Why did the author use 30 subjects for evaluation, it would be good to know the rationale behind the selection of using 30 subjects?
3. As, the small number of cases are used for the study, how do the author plan to generalize the findings to the larger and real population.
4. Can the author mention the steps that were taken to address the potential biases in the fictitious cases?
5. What were the prompts used by the authors, using the GPT-4 model to create realistic narrative for each fictitious case?
6. What were the prompts used by the text to audio model to generate the audio clips for the 30 subjects, this audio clips were used by the oncologist as a reference guide?
7. Section for Contextualization of the LLM Model, the customized GPT model was used to contextualize for toxicity classification. Can you elaborate on the process and steps used for the contextualization process and the types of information provided to the model?
8. Section for Analysis of results, Classification of the toxicities were classified into 3 groups, why was it grouped, and not calculated for each grade. As the grading provided initially will be redundant.
9. The authors use mode and mean accuracy to compare the LLM's classifications with those of oncologists. Can you provide more details on why these metrics were chosen and whether any additional metrics, such as precision, recall, or F1-score, were considered to evaluate the model's performance?
10. The results presented in this section were difficult to comprehend, it would have been good if authors would have provided a table with results for the general category and separate for the specific categories.
11. Section for types of errors, the classification of errors into mild and severe is a critical part of the analysis. It would be good, if authors could provide more specific examples of what constitutes mild versus severe errors, and how these errors could impact patient management in real-world setting?
12. The results show that the LLM's accuracy in general categories is comparable to that of oncologists. However, there is variability in specific categories. Could the authors provide result with standard deviation value to compare the variability.
13. Did the authors perform any statistical test for evaluating the performance of the LLM vs the oncologist evaluation? also, did the authors compare the results between multiple oncologists to see the agreement between the oncologist?
Comments on the Quality of English LanguageThe quality of English used in the manuscript was Good
Reviewer 3 Report
Comments and Suggestions for Authors
The manuscript “Leveraging Large Language Models for Precision Monitoring of Chemotherapy-Induced Toxicities: A Pilot Study with Expert Comparison and Future Directions” is an interesting read and adds to the body of work in generative artificial intelligence (genAI) for medical research. This work introduces a genAI method to classify subjective toxicities in 30 fictitious patients undergoing chemotherapy. The method introduced achieved an accuracy of 0.815 in general categories and 0.646 in specific categories. The authors then provided future directions for implementing AI in the healthcare setting. The manuscript is comprehensive, insightful, and an interesting read. It is deserving of publication after the following are addressed.
Major comments:
1. The number of citations in the introduction relating to LLM and its uses in the field of healthcare is lacking. Over the past two years, several articles have charted the direction of the use of genAI in healthcare. The authors should include these research articles in the introduction to give readers a comprehensive picture of the research landscape.
2. The authors mentioned that “the LLM used in this study was a version of OpenAI’s GPT-4, specifically contextualized for the task of toxicity classification”, and at several other instances in the manuscript, the term “contextualized LLM” is used. This raises two questions:
a. “A version” is too vague. There are several models of GPT-4, which one are the authors referring to?
b. Could the authors provide more details on how the contextualization was performed? Specifically, what kind of data and methodology were used to adapt GPT-4 for the task of toxicity classification?
c. If this contextualized LLM functions as a customized chatbot, what meta prompt was supplied to guide its responses? Understanding the meta prompt would offer insight into the parameters and instructions that were critical in shaping the model's behavior for toxicity classification.
3. This reviewer needs clarification about how the fictitious cases match up to actual clinical cases. Also, how did information change at each stage of generating these fictitious cases? An example of the results at each stage of the process would bring clarity to this.
4. In lines 684-685, the authors mention, “The accuracy achieved… falls within the range of accuracy of oncologist currently performing this function.” This statement is too vague and lacks the necessary statistical support. It would be more informative to provide a detailed evaluation, including specific accuracy metrics and confidence intervals, and compare the model's performance to that of oncologists using statistical tests. This would offer a clearer and more robust understanding of the model's effectiveness relative to human experts. At the very least, the authors should include a citation if this is studied in previous studies.
Minor Edits:
1. The section header for the Introduction should be in English.
2. There are too many bullet points in this article. Some lists (predominantly in the results section) only have a single item, which negates the need for a bullet point.
It is clear to this reviewer that the authors are experts in the field of toxicity monitoring; however, the authors should consider getting an expert in the field of LLM or genAI to read through the article to improve the sections describing the LLMs and to gain insights on how LLM research are reported. There is a stark difference in quality when comparing the parts describing your research in toxicity and the parts on genAI.
Reviewer 4 Report
Comments and Suggestions for Authors
This paper demonstrates the potential of Large Language Models (LLMs) in accurately classifying chemotherapy-induced toxicities, achieving performance levels comparable to expert oncologists. However, to strengthen the study further and provide a more comprehensive scientific contribution, I recommend the following revisions:
· Highlight specific gaps in the existing research and provide a concise statement of the main findings and recommendations in Abstract.
· The introduction could benefit from a more critical discussion of the existing literature. One of the topics that has recently attracted the attention of researchers is "transfer learning" from LLMs. Given the intense discussion around this issue in areas such as cancer genetics and mutation analysis, please consider.
· The methods section needs more detail on the creation of fictitious cases. Specify how the cases were designed, what criteria were used to ensure their realism, and how the expert knowledge was integrated into these designs.
· The distinction between mild and severe errors requires further elaboration. Discuss the clinical implications of the errors in more detail.
· Address the potential for bias in the LLM's training data and its impact on the results.
